# The Possibility of Intracranial Hypertension in Patients with Autism Spectrum Disorder Using Computed Tomography

**DOI:** 10.3390/jcm9113551

**Published:** 2020-11-04

**Authors:** Shuichi Yamada, Ichiro Nakagawa, Fumihiko Nishimura, Yasushi Motoyama, Young-Soo Park, Hiroyuki Nakase

**Affiliations:** Department of Neurosurgery, Nara Medical University, Kashihara, Nara 634-8521, Japan; nakagawa@naramed-u.ac.jp (I.N.); fnishi@naramed-u.ac.jp (F.N.); myasushi@naramed-u.ac.jp (Y.M.); park-y-s@naramed-u.ac.jp (Y.-S.P.); nakasehi@naramed-u.ac.jp (H.N.)

**Keywords:** autism spectrum disorder, intracranial hypertension, computed tomography, circularity, intracranial pressure

## Abstract

Although intracranial pressure is considered to be normal in children with autism spectrum disorder (ASD), we aimed to assess whether such children may have increased intracranial pressure using noninvasive computed tomography (CT). Head CT scans of children with ASD (109 cases, male 91 and female 18, average age 4.3 years) and of children with typical development (60 cases, male 35 and female 25, average age 4.5 years) were acquired. The images were processed to map the shape of the inner skull surface. We predicted that a complex skull shape, based on a marked digital impression, would be indicative of chronically increased intracranial pressure. The data of the scans were extracted and processed to automatically establish inner and outer cranial circumferences. The circularity (reflecting inner skull shape and area) and C-ratio (ratio of inner/outer circumference) were determined and statistically analyzed. The circularity and C-ratio were significantly lower in children with ASD than in children with typical development. A lower circularity was associated with a more complex shape of the inner skull surface, which indicated the presence of intracranial hypertension. Our study suggests that children with ASD may be at a risk for chronic intracranial hypertension. Our technique incorporating the circularity and C-ratio is a useful noninvasive method for screening such patients and could impact future investigations of ASD.

## 1. Introduction

Autism spectrum disorder (ASD) is a neurodevelopmental disorder with unclear etiology, but it has been considered an endogenous disease involving multiple factors [1,2]. Moreover, ASD is reportedly more likely to occur concomitantly with diseases associated with congenital cranial deformity such as craniosynostosis. An extrinsic component of an organic brain disorder has also been implicated in ASD development [3].

Craniosynostosis is associated with a chronic elevation of intracranial pressure (ICP), and the associated developmental disturbance in the brain has been implicated in the development of ASD [4,5,6]. There are some reports on the association between several neurological disorders and the development of ASD, particularly on the association between cranial or cerebral volume and ASD. There are reports of larger cerebral volume, larger ventricles, and larger head circumference than in typically developing cases [7,8,9,10,11,12,13], and most of these reports indicate that intracranial volume is larger than in typically developing cases. This could possibly suggest an elevation of ICP. However, there have been no reports that clearly demonstrate elevated ICP and its mechanism.

Furthermore, ICP sensor insertion is the gold standard method for measuring ICP [14,15]. However, this procedure is invasive and requires general anesthesia; hence, its application in such cases is controversial, given that the relationship between ICP and ASD remains unclear.

In the present study, we used a noninvasive method—computed tomography (CT)—to detect the presence of increased ICP in children with ASD and those with typical development. Given that many researchers believe that ICP is normal in children with ASD, our findings may be meaningful and could impact the direction of future ASD studies.

## 2. Methods

### 2.1. Participants

The study subjects included children diagnosed with ASD who were being treated for developmental disorders at Nara Medical University Hospital and Yamanobe Hospital consecutively for 2 years from January 2016 to December 2017; the control group comprised children with a normal skull shape. We selected children between the ages of 3 and 6 years as subjects. We recorded the age of the children at the time of the CT scan, rather than at the time of diagnosis. Children with congenital cranial shape abnormalities such as craniosynostosis and those with congenital syndromes, such as Down syndrome, were excluded. The flow chart for patient selection of the ASD group is shown in Figure 1. In almost all children, the diagnosis was made by specialists of ASD at a nearby pediatric psychiatric hospital, and the children were referred to our institution for rehabilitation-based treatment. The diagnosis of ASD and the assessment of its severity were based on the Diagnostic and Statistical Manual of Mental Disorders (DSM)-5.

Head CT scans were performed in the study group to screen for the presence or absence of intracranial organic lesions. The procedure and the risk for radiation were explained to the parents, and the procedure was conducted only after informed consent was obtained.

The control group comprised randomly selected children who had undergone CT for minor head trauma and were considered to be normal after an analysis by two radiologists and one neurosurgeon. Additionally, it was confirmed that they had no history of ASD based on a review of their medical records. Similar to the test group, the age of the subjects was recorded at the time of the CT scan. Accordingly, a total of 60 subjects were enrolled, including 15 subjects each aged 3, 4, 5, and 6 years, with no matching by age or sex with ASD children.

This study was conducted according to the guidelines of the Nara Medical University review board (authorization number 2381). The guidelines were compatible with the Declaration of Helsinki ethical principles for medical research.

### 2.2. Principles of Predicting Increased ICP

Digital impressions are alterations in the shape of the inner skull surface due to increased ICP. Their presence is assumed to reflect chronic increased ICP [16,17]. However, there is no established method to quantify the presence and degree of digital impressions. If digital impressions are present, the shape of the inner skull surface may be more complex than normal. Thus, we attempted to quantify this complexity using image processing.

One quantitative measure of the complexity of a graphic is its “circularity”, given by the following equation:
(1)$$C=4\pi \times \frac{S}{L^{2}}$$
(C = circularity; *S* = surface area; *L* = circumference).

Circularity is at its maximum value of 1 for circles, and additional complexity is associated with a lower value (e.g., 0.785 for squares and 0.604 for equilateral triangles). We speculated that the circularity would be lower if digital impressions were present, as these impressions would enhance the complexity of the inner skull surface. The circularity in the ASD group was statistically compared with that in the control group.

The outer circumference of the skull was also examined. Usually, the outer circumference is greater than the inner circumference, although this difference is expected to be smaller if digital impressions are prominent. Therefore, the ratio of the difference between the two circumferences to the outer circumference was also examined. This value can be calculated using the following formula:(2)$$C\;\mathrm{ratio}=\frac{Lout-Lin}{Lout}\times 100$$
(C-ratio = ratio of the difference between the outer and inner circumferences; *L-out* = outer circumference; *L-in* = inner circumference).

As with circularity, additional complexity of the inner skull shape is associated with a smaller C-ratio. We statistically compared the C-ratio between the ASD group and control group.

### 2.3. Image Processing Method for CT Data

CT (SCENARIA 64 v.1.92, HITACHI Ltd., Tokyo, Japan and Optimo 660CT, GE Healthcare, Chicago, IL, USA) was performed for both the ASD and control groups. The imaging conditions were as follows: non-helical CT, with a slice thickness of 5 mm and inclusion of the bone window. We used CT slices at the level of the third ventricle, as this slice lies just above the skull base with an innate complex morphology. Above this slice, the slice lies at an oblique angle to the skull, and, hence, it will be difficult to accurately capture the change in shape due to the digital impression at those levels (Figure 2a).

Only the bone information was extracted from the acquired CT data using Synapse Vincent (3D image analysis system, Fujifilm Corporation, Tokyo, Japan) (Figure 2b). The obtained data were used to create black and white images by tracing the inner and outer circumferences of the skull with image processing software (Photoshop Elements 15, Adobe Systems Incorporated, San Jose, CA, USA) (Figure 2c,d). The trace utilized the automatic selection function of the software. Images with clear internal and external circumferences were automatically processed using image measuring software (ImageJ, image processing and analysis open source soft, National Institutes of Health, Bethesda, MD, USA) to calculate both the length of the circumference and the surface area of the inner and outer circumferences. Using these values, the circularity and C-ratio were calculated. Figure 3 and Figure 4 show representative cases of subjects with ASD and subjects in the control group, respectively.

### 2.4. Analysis

The t-test was used to analyze the difference in sex and disease severity between groups, as these values followed a normal distribution. As the other measurements did not show a normal distribution, the non-parametric Mann–Whitney test was used for analysis instead. The IBM SPSS (International Business Machines Corp., Armonk, NY, USA) program was used for statistical analysis. The significance level was set at *p* < 0.05.

## 3. Results

A total of 241 patients were treated at our hospital during the study period. Data from 109 patients who met the inclusion criteria were analyzed. A total of 60 subjects, including 15 from each age group (3–6 years of age), were included in the control group. The characteristics of the ASD group and the control group are presented in Table 1. Table 2 describes the reasons for undergoing head CT scans in the control group.

The assessment of the severity was based on the Diagnostic and Statistical Manual of Mental Disorders (DSM)-5.

Circularity was significantly lower in the ASD group (Table 3), which implies that the shape of the intracranial surface was more complex—i.e., the prominence of the digital impression was greater in the ASD group than in the control group (cut-off value, 0.827). Examination by age group showed that circularity was significantly lower in patients aged <4 years, although no significant difference was observed in patients aged >5 years. The circularity values in the control group remained almost unchanged across age groups. However, in the ASD group, the values were slightly higher for the 5- and 6-year-olds. When categorized by sex, we found that circularity was significantly lower in boys. Although circularity tended to be low in girls as well, there was no significant difference in the values between the groups. The severity-categorized results showed a significant difference only for grade 1. Unexpectedly, the circularity values tended to be lower in milder cases of ASD.

The C-ratio was also significantly lower in the ASD group than in the control group (Table 4). When subjects were categorized by age and severity, significant differences in the C-ratio were observed for those aged ≤4 years and with severity grades of 1 and 2; similar findings were observed when the subjects were categorized by circularity. These results were significant for both boys and girls.

## 4. Discussion

The significance of brain volume in children with developmental disorders such as ASD remains controversial [12,13,18,19,20]. Most of the published studies have found that both brain volume and cranial volume are larger in children with ASD than in those included in control groups; some studies have conducted detailed analysis based on age, sex, body weight, and other variables and have yielded similar findings.

Although several reports have addressed the difference in brain volume in children with ASD, these reports have certain limitations. First, most of the studies were relatively small; their sample size was <100. Our findings may be considered more significant, as we included 109 samples. Second, none of the studies have assessed the relationship between brain volume and ICP. Although it is reasonable to believe that a greater brain volume would have certain effects on the ICP, none of the studies have assessed this hypothesis in greater detail. Third, various studies have proposed the mechanism underlying increased brain and cranial volume in children with ASD; however, none of these reports offer a definitive basis. Dysregulation of the hypothalamic–pituitary–adrenal axis [19], accelerated synaptic proliferation from the late fetal period to the early birth period [12], deceleration of the normal loss of neuronal processes, or overexuberant dendritic arborization are some of the potential explanations proposed [20], but these have limited scientific significance. Moreover, none of these mechanisms can explain the development of ASD, which is also true of the present study.

As none of these hypotheses can be considered to be of therapeutic significance, most researchers view that changes in skull volume and brain volume have little impact on clinical treatment. Nevertheless, in the present study, we found that chronic intracranial hypertension may be present in children with ASD.

Craniosynostosis may be one of the causes of such chronic intracranial hypertension [21,22]. Several reports have stated that developmental disorders frequently occur concomitantly in children with craniosynostosis [5,6]. We used a similar approach and found a significantly lower circularity value in children with craniosynostosis, as compared with those with a normal skull shape [23]. In addition, idiopathic intracranial hypertension has been reported as a cause of increased ICP in children [24,25], although no reports have addressed the relationship between this condition and digital impressions or ASD, possibly due to the relatively rapid increase in ICP. In the present study, we found that there is a reasonable possibility that some children with ASD might develop chronic intracranial hypertension. In children with similar ages as those in the study, digital impression might appear even under normal conditions; nevertheless, the differences in circularity or the degree of digital impression observed between children with ASD and the control group in our study appear to be very significant. We believe that the detection of these significant differences could be related to the degree of digital impression, which was converted to a numerical value; this enabled the detection of subtle differences in digital impressions that were difficult for even expert readers to detect visually.

ICP sensors have been conventionally placed intracranially to detect ICP, and they remain the gold standard approach [14,15]. However, this method is invasive and should not be used for screening. Recently, ICP measurement using magnetic resonance imaging (MRI) has been reported [26,27]. Unlike measurement with inserted ICP sensors, this method is noninvasive and revolutionary. Nevertheless, most of the subjects tested using this method include adolescents or adults. To obtain accurate data, the subjects need to remain still during the MRI. For children at an early age, the use of sedatives is likely to be necessary to keep them calm during the MRI procedure. In particular, in children with ASD, such a task might be very difficult without sedatives, which themselves present certain risks and complications [28,29]. Efforts have been made to measure ICP by attaching a sheet-like sensor to the forehead, although, at present, this method remains experimental [30]. In the present study, we used CT, which had an average imaging time of approximately 10 s. Even patients with ASD can be conveniently imaged in this short time period. In the present study, none of the patients required sedative use during imaging. Hence, one of the major benefits of this method is that repeated image-based evaluations can be conducted, as the burden on the patient is minimal.

Nevertheless, in the present study, we were unable to completely explain the findings based on severity. Our results show that circularity was significantly lower in patients with less severe disease. No consistent trend was observed for the C-ratio, but significant differences were noted for patients with grade 1 and grade 2 disease severity. Although greater severity was expected to result in lower circularity and lower C-ratio, the reverse was found to be true in our cases. However, there was no significant difference in the circularity and C-ratio in patients with grade 3 disease severity; in fact, the circularity tended to be lower than those in the control group. Furthermore, there was no significant difference in the number of patients with each grade. At present, we do not have a clear explanation for these results. A few reports have assessed the relationship between brain volume, head circumference, and disease severity. In particular, one such study of the relationship between brain volume, extra-axial fluid, and ASD severity showed that the ASD severity increased in proportion to the quantity of extra-axial fluid [31].

Furthermore, we found no significant difference in circularity in girls, although the value itself was lower in girls than in boys. However, this lack of significant difference might be attributed to the lower number of girls in the study cohort.

As mentioned above, the C-ratio analysis results were similar to the results of the circularity analyses. Nevertheless, we believe that the individual values are not proportional to the degree of ICP, even in categories where significant differences were observed due to wide variations in the maximum and minimum values. Hence, this finding needs to be assessed in another study with a larger sample size.

In the present study, we detected increased ICP in children with ASD. However, it remains a challenge to determine the true clinical application of this finding. Although we do not believe that surgical interventions, such as craniotomy, should be conducted in patients with intracranial hypertension based on these findings, we instead suggest that screening CT scans should be conducted in children with ASD to detect for the presence of intracranial hypertension. In particular, we believe that patients with lower cut-off values should be further examined for other symptoms of increased ICP with regular CT-based follow-up examinations or should be treated in collaboration with pediatric neurosurgeons from an early stage. Furthermore, we believe that our technique will be convenient and useful for long-term follow-up examinations.

### Limitations

The present study has certain limitations. First, it is necessary to examine whether the CT slice used for obtaining the circularity values was appropriately chosen. As mentioned previously, we analyzed the CT slice that was most likely to show digital impressions; however, it is difficult to determine whether this method is appropriate. Although the use of multiple slices may allow for a more accurate examination, it might be quite time-consuming and laborious. Similarly, the axial view, as well as the coronal and sagittal views, may better reflect the intracranial shape. The development of software that could automatically perform these tasks might represent an area for future research.

Second, the circularity values obtained in this study did not correlate with the absolute ICP. At present, absolute ICP can only be estimated via ICP sensor placement or lumbar puncture; however, it might be difficult to perform these tests in children with ASD. As mentioned previously, we conducted a similar study in children with craniosynostosis. ICP sensors were used to measure ICP preoperatively in some of these pediatric patients, but no correlation could be demonstrated because of the small number of patients.

## 5. Conclusions

The present study shows that digital impressions were more pronounced in children with ASD than in those included in the control group, which may suggest the presence of chronic elevated ICP. This significant novel finding suggests the presence of intracranial hypertension in children with ASD, and it could impact future investigations of ASD. Moreover, our method is noninvasive and can be used as a screening tool or for regular follow-up examinations with these patients.

## Figures and Tables

**Figure 1 jcm-09-03551-f001:**
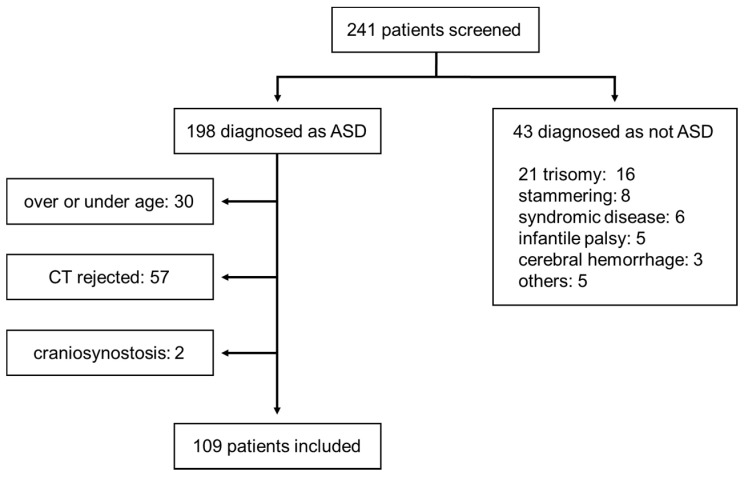
Selection of the study participants of the autism spectrum disorder (ASD) group.

**Figure 2 jcm-09-03551-f002:**
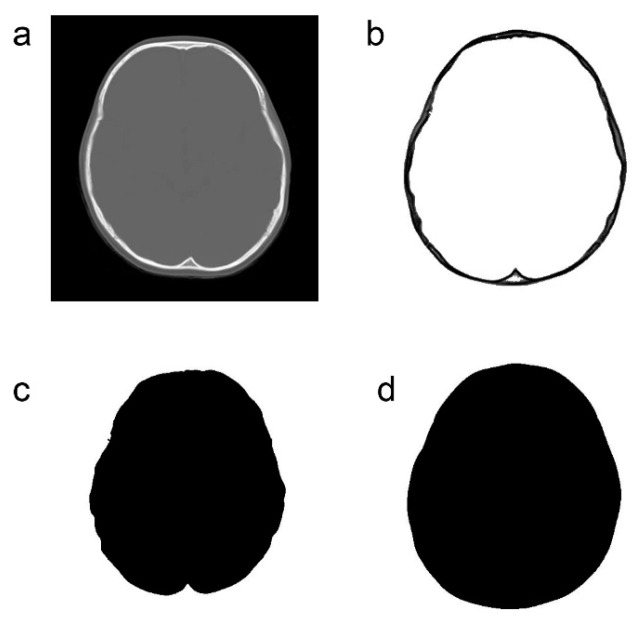
Image processing method for computed tomography data. (**a**): Original computed tomography image. (**b**): Extraction of only bone information. (**c**): Extraction of only the inner cranial circumference. (**d**): Extraction of only the outer cranial circumference.

**Figure 3 jcm-09-03551-f003:**
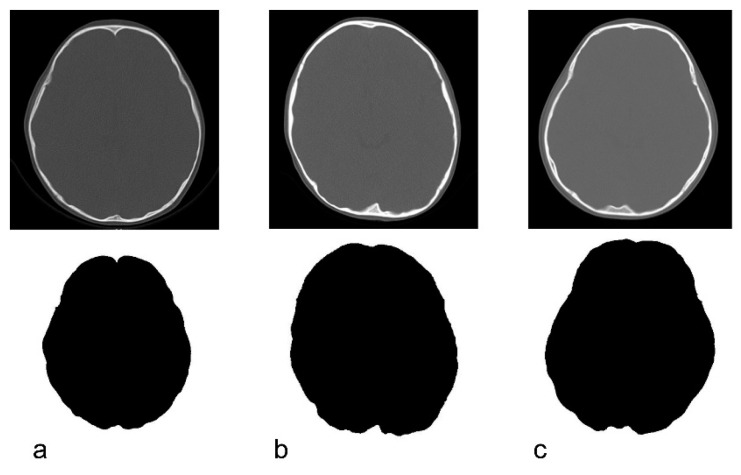
Original CT image and the depiction of the inner cranial circumference following image processing in children with autism spectrum disorder. (**a**): 3 years, circularity = 0.822. (**b**): 4 years, circularity = 0.824. (**c**): 5 years, circularity = 0.817.

**Figure 4 jcm-09-03551-f004:**
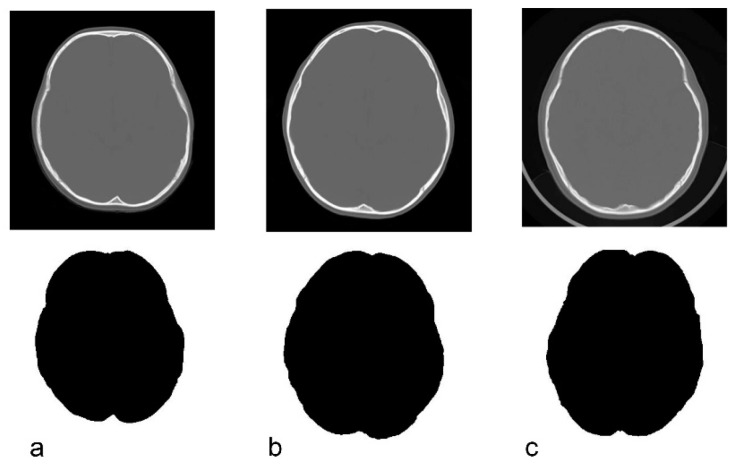
Original CT image and the depiction of the inner cranial circumference following image processing in normal control subjects. (**a**): 3 years, circularity = 0.845. (**b**): 5 years, circularity = 0.856. (**c**): 6 years, circularity = 0.844.

**Table 1 jcm-09-03551-t001:** Characteristics of the autism spectrum disorder group and the control group.

	ASD	Control
Average age (years)	4.3	4.5
Patient age		
3 years	27 (24.8%)	15 (25.0%)
4 years	39 (35.8%)	15 (25.0%)
5 years	27 (24.8%)	15 (25.0%)
6 years	16 (14.7%)	15 (25.0%)
Male sex	91 (83.5%)	35 (58.3%)
Severity		
Grade 1	37 (33.9%)	
Grade 2	32 (29.4%)	
Grade 3	40 (36.7%)	

**Table 2 jcm-09-03551-t002:** Reasons for undergoing computed tomography in the control group.

	No. (%)
Simple head injury	48 (80.0%)
Cerebral concussion	6 (10.0%)
Headache	2 (3.3%)
Syncope	2 (3.3%)
Febrile seizure	1 (1.7%)
Face injury	1 (1.7%)

**Table 3 jcm-09-03551-t003:** Comparison of circularity.

	Circularity		
	ASD	Control	*p*-Value
Average overall	0.822 ± 0.0235	0.831 ± 0.0190	0.021
By age			
3–4 years	0.822 ± 0.0238	0.831 ± 0.0225	0.048
5–6 years	0.823 ± 0.0231	0.831 ± 0.0153	0.191
By gender			
Male	0.823 ± 0.0229	0.833 ± 0.0178	0.027
Female	0.819 ± 0.0267	0.827 ± 0.0206	0.22
By severity			
Grade 1	0.820 ± 0.0215	0.831 ± 0.0190	0.01
Grade 2	0.823 ± 0.0200		0.062
Grade 3	0.824 ± 0.0277		0.21

Data are presented as mean ± standard deviation

**Table 4 jcm-09-03551-t004:** Comparison of the C-ratio.

	C-Ratio		
	ASD	Control	*p*-Value
Average overall	2.136 ± 1.349	2.703 ± 1.124	0.003
By age			
3–4 years	1.841 ± 1.435	2.430 ± 1.243	0.045
5–6 years	2.587 ± 1.072	2.976 ± 0.933	0.092
By gender			
Male	2.192 ± 1.341	2.781 ± 1.213	0.024
Female	1.849 ± 1.391	2.595 ± 0.999	0.023
By severity			
Grade 1	2.053 ± 1.195	2.703 ± 1.124	0.007
Grade 2	1.866 ± 1.453		0.002
Grade 3	2.427 ± 1.375		0.24

Data are presented as mean ± standard deviation

## Data Availability

The original data is available online at http://hdl.handle.net/10564/3596 and http://hdl.handle.net/10564/3597.

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
