# Peer review of "The Possibility of Intracranial Hypertension in Patients with Autism Spectrum Disorder Using Computed Tomography"

_jcm, 2020, doi:10.3390/jcm9113551_

Round 1

Reviewer 1 Report

Thank you for inviting me to review the paper entitled “The possibility of intracranial hypertension in patients with autism spectrum disorder using computed tomography”. This paper examines the head CT scans of children with ASD and typically developing children, finding that the head circularity and C-ratio were significantly lower in children with ASD than controls. Moreover, a lower circularity was associated with a more complex shape of the inner skull surface, which indicated the presence of intracranial hypertension.

I have appreciated the innovative and interesting topic, but I believe that this study has several flaws that would benefit from major revisions. General comments:

  1. The introduction is extremely synthetic and scarcely explicative of the study rationale
  2. The methodology is a little bit confused, in particular as concerns the participants recruitement and sample size
  3. The results are flawed by the absence of a correction for multiple comparisons

Below more specific comments and suggestions:

Abstract

  1. The authors could provide more information regarding the sample (i.e. number of participants with ASD and controls, age).
  2. I would suggest avoiding the expression “normal development”, but instead prefer “typically development” or “typically developing children”.

Introduction

  1. Introdusction --> Introduction
  2. “Autism spectrum disorder (ASD) is a congenital social disorder with controversial etiology,…”. I do not agree with this definition. ASD is not a “congenital social disorder”, but can be better defined as a “neurovelopmental disorder” causing impairments in socio-communication. Moreover the etiology is “unclear”, not “controversial”.
  3. Line 32: “An extrinsic component of an organic brain disorder has also 33 been implicated in ASD development”. Please, explain. The meaning of this sentence is really unclear.
  4. Line 37: “cases with ASD” is unclear. I suggest to use “individuals with ASD”
  5. Line 37: Please provide a reference to support this sentence and see my comment below.
  6. I think that the authors should expand the introduction. Apart from craniosynostosis there are other neurological disorders that might be associated with ASD. Moreover, what has been reported in literature regarding the association between ICP and ASD? Which are the findings of other researchers? This information is fundamental to get to the rationale fo the study, which otherwise will remain unclear.

Methods:

  1. Line 51-52:”as a more accurate ASD diagnosis is possible in this age group”. Not true. Please, delete.
  2. Figure 1: A further box with the number of participants with and without ASD would make the figure clearer.
  3. In general, from the methods, it is not clear how many children with ASD were recruited.
  4. Please, clarify selection procedure and the number of participants enrolled. How were the controls selected? 60 on the basis of what? Consecutively selected? Matched by age or sex? Please, explain
  5. Were children with ASD and typically developing children matched by age and/or sex?

Results

  1. Table 1: why did you present only the characteristics of the ASD group? Please, add socio-demographic information of the control group as well
  2. Table 4 and Figure 6: Please delete “normal”
  3. Table 3 and Figure 5 / Table 4 and Figure 6 à these pairs of tables/figure repeat the same data! Please, choose figures OR tables to avoid redundancy. Moreover, in the figures there are some errors in the bars regarding the age group (<3 and >=3), while there should be 3-4 and 5-6.
  4. In case you chosse to keep the Figures, please add the means and SD, as well as the p-values in the figures.
  5. I suggest to correct for multiple comparisons (e.g. Bonferroni’s correction or Benjamini-Hochberg Procedure)

Reviewer 2 Report

Authors have written a comprehensive article with easy to understand English and it flows smoothly explaining background, methodology, results and discussion.

My specific comments are:

  • Page 2 line 64: control group comprised of children who had undergone CT head for minor head trauma. How could the authors be certain that children in control group could not have autism?
  • Page 2 line 68: authors have followed their institutional guidelines. Kindly clarify the process of ethics approval as children in this study have been exposed to CT scan that is additional radiation risk. This may not be clinically relevant in author’s local area but is important for rest of the world. Are the institutional guidelines compatible with declaration of Helsinki ethical principles for medical research? If authors propose taking a single slice of CT scan it is still relevant to mention the radiation risk.
  • Table 1: clarify how the ASD is graded into 1, 2 and 3.
  • Table 3 and Figure 5 show similar results, do you need both?
  • Table 4 and figure 6 show same results, choose to keep either / or.

Discussion

  • Page 9, 202-207: You have stated that digital impression was significantly different compared to controls in your study. Could you explain why this was not significant for the Grade 3 patients? The same pathology that causes mild to moderate problem is not relevant for severe cases.
  • Page 10, line 244- 249: Conclusions from your study are primitive to conclude anything beside that this study opens up a possibility of treatment potential for ASD patients and develop understanding. What do you think about medical treatment of raised ICP?
